# Vitamin A Promotes the Fusion of Autophagolysosomes and Prevents Excessive Inflammasome Activation in Dextran Sulfate Sodium-Induced Colitis

**DOI:** 10.3390/ijms24108684

**Published:** 2023-05-12

**Authors:** Hiroto Hiraga, Daisuke Chinda, Takato Maeda, Yasuhisa Murai, Kohei Ogasawara, Ryutaro Muramoto, Shinji Ota, Keisuke Hasui, Hirotake Sakuraba, Yoh Ishiguro, Shukuko Yoshida, Krisana Asano, Akio Nakane, Shinsaku Fukuda

**Affiliations:** 1Department of Gastroenterology and Hematology, Hirosaki University Graduate School of Medicine, Hirosaki 036-8562, Japan; hhiraga@hirosaki-u.ac.jp (H.H.); dragon_0819_0327@yahoo.co.jp (R.M.);; 2Division of Endoscopy, Hirosaki University Hospital, Hirosaki 036-8563, Japan; 3Division of Gastroenterology and Hematology, Hirosaki National Hospital, National Hospital Organization, Hirosaki 036-8545, Japan; 4Shibata Irika Co., Ltd., Hirosaki 036-8084, Japan; 5Department of Microbiology and Immunology, Hirosaki University Graduate School of Medicine, Hirosaki 036-8562, Japan

**Keywords:** vitamin A, retinoic acid, pyroptosis, autophagy, inflammatory bowel disease

## Abstract

Vitamin A ensures intestinal homeostasis, impacting acquired immunity and epithelial barrier function; however, its role in innate immunity is mostly unknown. Here, we studied the impact of vitamin A in different dextran sulfate sodium (DSS)-induced colitis animal models. Interestingly, more severe DSS-induced colitis was observed in vitamin A-deficient (VAD) mice than in vitamin A-sufficient (VAS) mice; the same was observed in VAD severe combined immunodeficient mice lacking T/B cells. Remarkably, IL-1β production, LC3B-II expression, and inflammasome activity in the lamina propria were significantly elevated in VAD mice. Electron microscopy revealed numerous swollen mitochondria with severely disrupted cristae. In vitro, non-canonical inflammasome signaling-induced pyroptosis, LC3B-II and p62 expression, and mitochondrial superoxide levels were increased in murine macrophages (RAW 264.7) pretreated with retinoic acid receptor antagonist (Ro41-5253). These findings suggest that vitamin A plays a crucial role in the efficient fusion of autophagosomes with lysosomes in colitis.

## 1. Introduction

Vitamin A, a fat-soluble vitamin, is the precursor of retinoic acid, which, in turn, participates in several biological processes involved in vision, reproduction, epithelial differentiation, bone development, and immunity [1,2]. Vitamin A is present in foods in the form of retinyl ester, and provitamin A carotenoid is stored as a retinyl ester derivative in the stellate cells of the liver. Vitamin A is transported to target cells in the form of retinol and acts via nuclear receptors called retinoic acid receptor (RAR) and retinoid X receptor (RXR) [3,4,5].

Vitamin A deficiency is one of the major nutritional deficiency syndromes in developing countries and is associated with an increased incidence of infectious diseases [6]. Vitamin A deficiency caused a reduction in α4β7+ memory/activated T cells in the lymphoid organs and the depletion of T cells from the intestinal lamina propria (LP) in mice [7]. In addition, vitamin A deficiency was shown to impair innate immunity, impeding the normal regeneration of mucosal barriers damaged by infection [8,9] and diminishing the function of neutrophils [10,11], natural killer cells [8,12], and macrophages. Notably, macrophages, the major population of tissue-resident mononuclear phagocytes, play a key role in bacterial recognition and elimination as well as in the polarization of innate and adaptive immune responses. Importantly, although vitamin A deficiency promotes the production of interleukin (IL)-12 and IL-1 by macrophages, their phagocytic capacity is impaired [13].

The precise etiology of inflammatory bowel diseases (IBD), including Crohn’s disease (CD) and ulcerative colitis (UC), remains unclear; however, it is well documented that macrophages are deeply involved in the onset and development of IBD [14]. In the 1980s, vitamin A therapy in patients with CD was found to be insufficient to sustain remission [15]. However, the use of Elental, an elemental diet for patients with CD containing high levels of vitamin A (600 IU/pack; adult requirements: 1800–2000 IU/day), correlated with an imbalance between pro-inflammatory and anti-inflammatory responses in CD [16].

Pyroptosis is inflammasome-induced lytic cell death and plays important roles in anti-bacterial innate immune defenses and lethal endotoxemia; however, its role in chronic inflammatory diseases is unknown. Cytoplasmic caspase-11 (or caspase-4 and -5 in humans) is closely related to the non-canonical inflammasome and pyroptosis in the context of Gram-negative bacterial agents [17,18,19,20]. Lipopolysaccharide (LPS) is the bacterial pathogen-associated molecular pattern that promotes the activation of caspase-11 [21,22]. The proteolytic cleavage of gasdermin D (GSDMD) by mouse caspase-11 or human caspase-4 and -5 is essential for pyroptosis in macrophages in the context of LPS [23]. Moreover, caspase-11 triggers the NLR family pyrin domain-containing 3 (NLRP3)- and adopter protein apoptosis-associated speck-like protein containing a CARD (ASC)-dependent activation of caspase-1, resulting in the proteolytic maturation of the inactive cytokines pro-IL-1β and pro-IL-18 [22,24,25]. Importantly, cytoplasmic caspase-1 is associated with the canonical inflammasome activated by the NLRP3-mediated recognition of danger signals; therefore, caspase-11 also serves as a bridge between non-canonical and canonical inflammasomes. Recently, Evavold et al. reported that multiple microbial and self-derived stimuli could induce IL-1 release from living macrophages under the recognition of the pore-forming protein GSDMD [26]. Considering inflammatory and immune responses, IL-1 is a key mediator released by monocyte-derived macrophages. Interestingly, the level of IL-1 in the colonic mucosa was significantly elevated during active IBD [27]. In addition, upregulated levels of caspase-4 and -5 mRNA have been detected in biopsy specimens from patients with active IBD [28]. However, the pathological role of IL-1, caspases, and pyroptosis in colitis remains unclear.

Macroautophagy (abbreviated as “autophagy”) represents a homeostatic cellular mechanism essential for the turnover of organelles and proteins through lysosome-dependent degradation processes. The variant ATG16L1, an ortholog of the gene encoding the yeast autophagy-related protein 16 (ATG16), has been associated with CD [29], suggesting that the inability of autophagy might account for some types of CD. This hypothesis is further supported by the increased secretion of IL-1β and IL-18 in LPS-treated ATG16L1 or ATG7 knockout macrophages, as well as the increased susceptibility of ATG16L1 knockout mice to dextran sulfate sodium (DSS)-induced colitis, which could be ameliorated by the injection of IL-1β and IL-18 blocking antibodies [30]. Importantly, cytokine activation in response to LPS and adenosine triphosphate (ATP) in wild-type macrophages is dependent on the NLRP3 inflammasome pathway. Notably, NLRP3 inflammasome activation has been associated with autophagy deficiency and consequent mitochondrial dysfunction, including the enhanced production of mitochondrial reactive oxygen species (ROS), and increased mitochondrial membrane permeability [30,31]. Accordingly, these observations suggest that autophagy dampens the activation of the inflammasome pathway via the stabilization of mitochondria and the maintenance of mitochondrial quality control processes.

Familial Mediterranean fever (FMF) is the most common monogenic auto-inflammatory disease, with a particularly high prevalence of 1:500 to 1:1000 among individuals of Middle Eastern and Mediterranean descent [32]. FMF is caused by missense mutations in the MEFV gene, which are critical for activating the pyrin inflammasome. Although the role of MEFV in human disease remains poorly understood, recent data suggest that the FMF-associated MEFV variants are gain-of-function mutations resulting in increased responsiveness to bacterial products. Mice bearing the FMF-associated MEFV mutations exhibited more efficient caspase-1 activation than their wild-type counterparts, resulting in IL-1β overproduction in response to LPS alone [33]. Recent studies suggest an association between MEFV and IBD. Sharma et al. reported that pyrin (MEFV) is required for inflammasome activation and IL-18 maturation [34]. Moreover, Arasawa et al. reported that colonic lesions mimicking CD can be observed in FMF [35].

Herein, we used different DSS-induced colitis animal models to elucidate the role of vitamin A (focusing on the metabolite retinoic acid) in regulating inflammasome signaling and pyroptosis. Overall, our data show the importance of vitamin A and highlight retinoic acid as a potential new therapeutic approach to treat IBD and possibly other auto-inflammatory diseases.

## 2. Results

### 2.1. Vitamin A-Deficient (VAD) Mice Exhibit More Severe DSS-Induced Colitis Independent of T and B Cells

Vitamin A is a key mediator of intestinal homeostasis and immunity; however, the impact of vitamin A on gut inflammation remains poorly understood. Our data revealed that serum vitamin A levels in patients with active CD were significantly lower than those in patients with inactive CD (Appendix A). Moreover, serum vitamin A levels in patients with CD significantly correlated with C-reactive protein (CRP) (Appendix A). Therefore, to understand the role of vitamin A in gut inflammation, we investigated the effect of vitamin A deficiency on murine experimental colitis. We used DSS to induce colitis in vitamin A-deficient (VAD) and vitamin A-sufficient (VAS) C57BL/6J (B6) mice. Importantly, VAD B6 mice developed a more severe colitis phenotype than VAS mice, considering the survival rate (Figure 1b), % body weight (Figure 1c), colonic length (Figure 1a,d), and histological features (Figure 1e,f). Previously, a study using C.B-17 severe combined immunodeficient (SCID) mice showed that acute DSS-induced colitis does not require the presence of T or B cells [36]. Herein, to confirm that T and B cells are indeed not essential for the exacerbation of DSS-induced colitis in VAD mice, 2.5% DSS was used to induce colitis in VAD and VAS C57BL/6J SCID mice lacking T and B cells. Similar to B6 mice, VAD SCID mice showed a more severe colitis phenotype than VAS SCID mice, as evidenced by the survival rate (Figure 2b), % body weight (Figure 2c), colonic length (Figure 2a,d), and histological findings (Figure 2e,f). Altogether, these results indicate that vitamin A deficiency promotes the exacerbation of colitis in a T- and B-cell-independent fashion.

### 2.2. Inflammasome-Related Protein Levels Increase in the Colonic LP of VAD Mice

Given that the LP comprises various hematopoietic and endothelial cells that modulate innate immunity, we next quantitated cytokine production in the LP of DSS-treated VAS and VAD B6 mice using a multiplex protein-based system. Remarkably, the colonic LP of VAD B6 mice showed high levels of IL-1β (Figure 3a) and IL-18 (Figure 3b) before DSS treatment but not of tumor necrosis factor (TNF)-α (Figure 3c). Moreover, the colonic LP of VAD B6 mice displayed increased protein levels of caspase-11, GSDMD, NLRP3, ASC, caspase-1, and IL-1β, both in the steady state and following DSS treatment (Figure 3d). Additionally, caspase-1 activity in the colonic LP of DSS-treated VAD B6 mice was significantly increased when compared with that of VAS B6 mice (Figure 3e). Overall, these results suggest that the excessive production of IL-1β and IL-18 following upregulated activation of inflammatory caspases in the LP could explain the exacerbation of VAD DSS-induced colitis.

### 2.3. Pyroptosis via Non-Canonical Inflammasome Signaling Increases in Macrophages Pretreated with a RAR Antagonist

Patients with IBD exhibit elevated IL-1 levels in freshly isolated LP mononuclear cells compared with that in control subjects [37]. Moreover, Pizarro et al. reported that IL-18, which is primarily produced by intestinal epithelial cells, tissue macrophages, and dendritic cells, is upregulated in patients with CD [38]. Herein, we focused on macrophages as the source of IL-1β and IL-18 in the LP. First, as an in vitro system of our VAD model, we used macrophages (RAW 264.7 cells) in the presence or absence of a RAR antagonist, followed by incubation in the presence of LPS, and examined pyroptosis. Notably, pyroptosis is distinguished from other forms of cell death by unique morphological and biochemical characteristics. Importantly, the viability of macrophages pretreated with the RAR antagonist was significantly reduced when compared with that of the corresponding macrophages pretreated with the vehicle (Figure 4a). Following LPS treatment, the viability of macrophages pretreated with the RAR antagonist was considerably lower than that of control macrophages (Figure 4b). The supernatants of macrophages pretreated with the RAR antagonist were substantially alkaline, reflecting the decrease in cell viability (Figure 4c). Importantly, the macrophages pretreated with the RAR antagonist showed morphological features suggestive of pyroptosis, such as cell swelling, cell lysis, and the release of cellular contents, regardless of LPS treatment (Figure 4d–g). Moreover, the macrophages pretreated with the RAR antagonist released high amounts of lactate dehydrogenase (LDH), an indicator of cell membrane damage, regardless of LPS treatment (Figure 4h,i). The secretion of IL-1β and IL-18 into the culture medium was also increased in RAR antagonist-pretreated macrophages, especially following LPS treatment (Figure 4j,k). However, TNF-α levels were unaltered in the examined groups (Figure 4l). Similar findings were detected in RAR antagonist-pretreated macrophages stimulated with peptidoglycan (PGN) and flagellin (Appendix A). Consistent with the above results, RAR antagonist-pretreated RAW 264.7 cells displayed elevated levels of (active) cleaved caspase-11 (30 kDa), even in the absence of LPS stimulation (Figure 4m). Cell lysates of RAR antagonist-pretreated RAW 264.7 cells exhibited increased levels of IL-1β in response to LPS (Figure 4m). We also used macrophages collected from our VAD model (CD11b^+^ splenic cells and peritoneal exudate cells (PECs)) to perform ex vivo assays. Similar to the observed findings in RAW 264.7 cells pretreated with the RAR antagonist, LPS-treated CD11b^+^ splenic cells collected from VAD mice produced markedly higher levels of IL-1β and IL-18, but not of TNF-α, than counterpart macrophages collected from VAS mice (Appendix A). In contrast, PECs collected from both VAD and VAS mice pretreated with the RAR antagonist produced lower IL-1β levels than control cells after LPS treatment (Appendix A). These results suggest that pyroptosis, through non-canonical inflammasome signaling, is enhanced in macrophages in the absence of retinoic acid-mediated signaling.

### 2.4. Vitamin A Deficiency Decreases Autophagic Flux via Reduced Fusion of Autophagosomes and Lysosomes

Recent observations have revealed a relationship between autophagic proteins and the maturation of inflammasome-associated pro-inflammatory cytokines in macrophages [30,31,39]. In line with these findings, we examined autophagy in the context of vitamin A deficiency using three different methods. First, immunoblotting of autophagy-related proteins was performed in colonic LP tissues collected from VAS and VAD mice two days after DSS treatment (Figure 5a). VAD mice with DSS-induced colitis displayed equal levels of p62 but higher levels of LC3-II than VAS mice with DSS-induced colitis. Next, to assess autophagosome formation, colon samples were collected from VAS and VAD GFP-LC3 transgenic mice (GFP-LC3#53) 24 h after DSS treatment (Figure 5b). Interestingly, no GFP-LC3 signals were detected in the colon of VAS mice. In contrast, the colon of VAD mice exhibited diffuse GFP-LC3 signals in the LP with a few punctate dots. Moreover, electron microscopy analysis of the colonic LP of VAS and VAD mice was performed to quantify the number of autophagosomes. Interestingly, the LP mononuclear cells (LPMC) of VAD mice showed a greater abundance of swollen mitochondria with severely disrupted cristae than those of VAS mice, both in the steady state and following DSS treatment (Figure 5c). Additionally, to address the impact of retinoic acid on the autophagic flux of macrophages, we used immunoblotting to detect levels of autophagy-related proteins in RAW 264.7 cells (Figure 5d). RAW 264.7 cells were pretreated with or without 10 mM of the RAR antagonist (Ro41-5253) for 1 h, stimulated with or without 10 µg/mL LPS for 24 h, and then harvested. Compared with vehicle-treated macrophages, RAR antagonist-treated macrophages displayed elevated levels of both p62 and LC3-II in the absence of LPS. Conversely, although the levels of p62 remained unchanged, LPS treatment could decrease levels of LC3-II in both RAR antagonist-treated and untreated macrophages. Furthermore, we observed that RAR antagonist-treated macrophages generated high amounts of mitochondrial superoxide anion radical (O-) (detected by MitoSOX fluorescence) after 4 h of incubation in Earle’s Balanced Salt Solution (EBSS) medium (Figure 5e), suggesting that vitamin A deficiency is associated with increased mitochondrial ROS production and mitochondrial dysfunction. Overall, these results suggest that vitamin A deficiency decreases the autophagic flux of macrophages by reducing the fusion between autophagosomes and lysosomes.

### 2.5. VAD Impairs the Autophagic and Listericidal Activities of Macrophages, Promoting Increased Susceptibility to Listeria Monocytogenes Infection

*Listeria monocytogenes* is a Gram-positive bacterium that causes sepsis and meningitis in immunocompromised hosts and severe fetal infection during pregnancy. *L. monocytogenes* reaches the safe intracellular niche by interacting with proteins from the internalin family on the surface of host cells (usually macrophages), which is followed by phagocytosis [40,41,42]. Importantly, within macrophage vacuoles, *L. monocytogenes* bacteria secrete listeriolysin O (LLO), which lyses the vacuolar membrane and activates nuclear factor-κB (NF-κB)-mediated transcription of innate immune-response genes, such as CC-chemokine ligand 2 (CCL2). Owing to these characteristics, *L. monocytogenes* is widely used to analyze macrophages in vivo and in vitro. Accordingly, we investigated the effect of vitamin A deficiency in a mouse model of *L. monocytogenes* systemic infection. VAS and VAD B6 mice were infected intravenously with 5 × 10^5^ colony-forming units (CFU) of viable *L. monocytogenes*, and their survival was recorded until day 10 post-infection; the bacterial numbers in the organs were also determined 1 and 2 days after infection. Importantly, while only 17% of VAS B6 mice died during the study period, VAD B6 mice demonstrated a mortality rate of 92% (Figure 6a). Moreover, VAD mice exhibited significantly higher spleen bacterial numbers than VAS B6 mice (Figure 6b,c). Additionally, to ensure that this phenotype was not mediated by T and B cells, we repeated the experimental approach using C57BL/6J SCID mice. Similar to the findings in B6 mice, VAD SCID mice had significantly higher spleen bacterial numbers than VAS SCID mice (Figure 6b,c). Collectively, these data suggest that vitamin A is beneficial in the early stage of *L. monocytogenes* infection and that both T and B cells are not involved in the exacerbation of *L. monocytogenes* infection in VAD mice. Next, we focused on macrophages as the principal mediators of *L. monocytogenes* killing. A previous study has shown that vitamin A deficiency can impact the phagocytic and bactericidal activities of peritoneal macrophages in *Staphylococcus aureus* infection [43]. Herein, we evaluated the phagocytic activity and listericidal potential of splenic macrophages in the presence of vitamin A deficiency ex vivo. As expected, both phagocytic and listericidal activities of splenic macrophages derived from VAD mice were significantly lower than those of VAS mice (Figure 6d). Recently, autophagy was implicated in limiting the replication of *L. monocytogenes* in vivo [44,45]. Therefore, we next examined autophagosome formation during *L. monocytogenes* infection using liver samples from VAS and VAD GFP-LC3#53 mice, two days after infection with 5 × 10^5^ CFU of DsRedEx-labeled *L. monocytogenes* (Figure 6e). Although few autophagosomes and *L. monocytogenes* were observed in the livers of VAS GFP-LC3#53 mice, a clear co-localization of *L. monocytogenes* and autophagosome-derived signals could be observed in the livers of VAD GFP-LC3#53 mice. This observation suggests an inefficient autophagolysosomal maturation in VAD GFP-LC3#53 mice during *L. monocytogenes* infection. Finally, VAS and VAD B6 mice spleens were subjected to electron microscopy analysis 24 h after infection with 5 × 10^5^ CFU of *L. monocytogenes* (Figure 6f). Consistent with the above results, the spleens of VAD B6 mice showed more autophagosomes in the steady state and more autophagosomes with intact bacteria following *L. monocytogenes* infection than those of VAS B6 mice. Overall, our results indicate that VAD impairs the autophagic and listericidal activities of macrophages, thereby promoting increased susceptibility to *L. monocytogenes* infection.

### 2.6. Blockade of IL-1β Ameliorates DSS-Induced Colitis in VAD Mice

To date, no clinical trial has explored the potential administration of monoclonal antibodies (mAb) to directly target IL-1 in patients with IBD. It has been reported that the condition of patients with CD [46] and UC [47] worsens after administering anakinra (IL-1 receptor antagonist). Herein, we aimed to assess the impact of mAb against IL-1β on DSS-induced colitis in VAD B6 mice (Figure 7a,b). VAD B6 mice were administered 50 mg/day of anti-IL-1β mAb intraperitoneally from day 0 (when DSS treatment was initiated) to day 3; mice were fed 4% DSS and were followed up until day 10. Mice treated with anti-IL-1β mAb demonstrated a marked reduction in disease severity, as determined by the survival rate (Figure 7a) and body weight loss (Figure 7b). These results indicate that IL-1β contributes to the exacerbation of DSS-induced colitis in VAD mice. We next speculated whether the increased IL-1β level in the colonic LP and the consequent colitis exacerbation observed in VAD mice are dependent on canonical or non-canonical inflammasome signaling. To address this question in vivo, we used a chemical NLRP3 inhibitor, MCC950. VAD B6 mice were administered 10 mg/kg/day of MCC950 intraperitoneally starting on day 0, followed by every two days thereafter, and were fed 4% DSS (Figure 7c,d). Interestingly, disease severity was unchanged in MCC950-administered mice, as demonstrated by their survival rate (Figure 7c) and body weight loss (Figure 7d). Therefore, this result suggests that canonical inflammasome signaling does not contribute to the exacerbation of DSS-induced colitis in VAD mice. Additionally, we used a potent and specific mTOR inhibitor, rapamycin, to determine whether the promotion of autophagosome formation can prevent the exacerbation of DSS-induced colitis in VAD mice. VAD B6 mice were intraperitoneally administered 10 mg/kg/day of rapamycin daily from day 0 to day 7 and fed 4% DSS (Figure 7e,f). Rapamycin-treated mice showed increased disease severity, as determined by the survival rate (Figure 7e), although body weight loss was not observed (Figure 7f). These results suggest that the impairment of autophagolysosomal maturation in VAD mice was aggravated owing to excessive autophagosome formation following rapamycin administration.

## 3. Discussion

In the present study, we showed that retinoic acid could prevent excessive inflammasome signaling and consequent pyroptosis by promoting autophagic flux. VAD mice demonstrated severe experimental colitis and increased expression of inflammasome-related proteins (caspase-11, GSDMD, NLRP3, ASC, caspase-1, IL-1β, and IL-18) in the colonic LP. Moreover, the LPMC of VAD mice exhibited a considerable abundance of swollen mitochondria with severely disrupted cristae, indicating diminished autophagy. Additionally, murine macrophages pretreated with a RAR antagonist showed excessive pyroptosis due to non-canonical inflammasome signaling following autophagic dysfunction and diminished autophagosome fusion with lysosomes in vitro. Furthermore, we demonstrated that vitamin A deficiency could promote susceptibility to *L. monocytogenes* infection. Importantly, the blockade of IL-1β ameliorated experimental colitis in VAD mice.

Although there has been considerable discussion regarding the importance of vitamin A primarily in infectious diseases, its role in intestinal inflammation and the relationship between vitamin A and inflammasome signaling and autophagy are largely unknown. Vitamin A deficiency can reportedly exacerbate experimental colitis in BALB/c mice [48] and rats [49]. Consistent with the hypothesis that vitamin A prevents acute experimental colitis, BALB/c mice administered all-trans retinoic acid, an active metabolite of vitamin A, were found to be resistant to DSS-induced colitis [50]. Although activated caspase-1 is crucial for DSS-induced inflammation, caspase-1- or NLRP3-deficient mice developed significantly less severe colitis than wild-type mice. Importantly, this phenotype could be attributed to reduced IL-1β release from macrophages [51]. In addition, MCC950, a chemical NLRP3 inhibitor, can significantly suppress the release of the pro-inflammatory cytokines IL-1β, IL-18, and IL-1α in colitis [52,53]. However, recent studies have suggested that the NLRP3 inflammasome is important for maintaining gut homeostasis and has a protective role in intestinal inflammation. For instance, Zaki et al. reported that NLRP3 deficiency in mice led to a loss of epithelial integrity, resulting in the translocation of commensal bacteria, massive leukocyte infiltration in the colon, and more severe DSS-induced colitis [54]. The authors also revealed that protection was mediated via IL-18 secretion, with an injection of exogenous recombinant IL-18 partially alleviating the inflammatory symptoms of DSS-induced colitis. However, IL-18 was also found to induce interferon (IFN)-γ and promote Th1 responses. Early reports indicated that IL-18 was upregulated in patients with IBD, leading to upregulated levels of pro-inflammatory cytokines such as TNF-α, IL-1, and IL-6 [38,55]. These contradictory results could be attributed to the multifunctional role of IL-18 in colitis. In this respect, it could be concluded that the equilibrium between epithelial IL-18 and LP-produced IL-18 is critical for barrier function in colitis. Epithelial IL-18 may be beneficial, while IL-18 produced by LP cells may be detrimental to barrier function in colitis. To support this hypothesis, further comparison of IL-18 expression in the epithelial fraction and the colonic LP of VAD B6 mice is required.

The non-canonical inflammasome, comprising murine caspase-11 (or the human orthologs caspase-4/5) and GSDMD, has a protective role against Gram-negative bacterial infection; however, little is known regarding its function in colitis. Demon et al. confirmed that caspase-11 was constitutively expressed in the colon, and caspase-11-deficient mice were hyper-susceptible to DSS-induced colitis [56]. Furthermore, Oficjalska et al. have shown that impaired IL-18 production in caspase-11-deficient mice could result in reduced intestinal epithelial proliferation and increased cell death [57]. Considering GSDMD, Bulek et al. reported that GSDMD expression was increased in biopsies and isolated intestinal epithelial cells from the gut mucosa of patients with IBD when compared with healthy controls. Moreover, GSDMD-deficient mice developed substantially less severe DSS-induced colitis than healthy controls [58]. In contrast, Ma et al. have shown that GSDMD is highly activated in the intestines of DSS-treated mice and suggested that GSDMD deficiency would exacerbate DSS-induced colitis [59]. These contradictory results may be explained, at least in part, by environmental factors, including the microbiota, similar to those observed in caspase-11-deficient mice. Herein, we report that VAD mice exhibited severe DSS-induced colitis associated with increased caspase-11 and GSDMD in the colonic LP, similar to that noted in patients with IBD. One possible mechanism underlying this association may be related to increased epithelial permeability in VAD mice, resulting in bacterial translocation into the colonic LP. Additionally, LPMCs of VAD mice may be prepared for inflammasome activation in the steady state. Therefore, excessive inflammasome activation increases when DSS treatment compromises the epithelium. Bacterial numbers in the spleens of VAD SCID mice infected intravenously with *L. monocytogenes* were significantly higher than those of VAS SCID mice. Therefore, considering the above points, we believe that increased epithelial permeability is partially responsible for the hyper-susceptibility to DSS-induced colitis in VAD mice.

Several studies have demonstrated that autophagy dysfunction can lead to abnormal inflammasome activation, which, in turn, is associated with increased caspase-1 activity, elevated IL-1β and IL-18 production, and enhanced susceptibility to experimental intestinal inflammation in mice [60,61]. Caspase-11 (caspase-4) was shown to promote autophagosome formation in response to intracellular pathogens [62]. In the present study, immunoblotting, light microscopy, and electron microscopy data suggest that VAD mice showed decreased autophagic flux due to the diminished fusion between autophagosomes and lysosomes. Correspondingly, murine macrophage in vitro cultures mimicking the VAD in vivo model presented excessive pyroptosis through non-canonical inflammasome activation following dysfunction of autophagy due to the diminished fusion between autophagosomes and lysosomes and consequent increase in mitochondrial ROS production. These results are in line with those of macrophages with depletion of the autophagy gene *Map1lc3b* (microtubule-associated protein 1 light chain 3B), which exhibited dysfunctional mitochondria associated with increased ROS production and consequently increased secretion of IL-1β and IL-18 in response to LPS [31].

In addition, VAD mice showed increased susceptibility to *L. monocytogenes* infection, which is also associated with diminished autophagy in macrophages. Consistently, Tan et al. have reported that the expression of the CD-associated *ATG16L1^T300A^* risk allele led to a defect in plasma membrane damage repair upon *L. monocytogenes* infection and increased cell-to-cell bacterial spread [63]. Furthermore, the IBD susceptibility gene *GPR65* (coding for the G protein-coupled receptor 65, an H^+^-sensing G protein-coupled receptor) plays a role in maintaining the lysosomal function, preserving autophagy and colitis risk [64]. Considering these observations, vitamin A plays a key role in the efficient fusion between autophagosomes and lysosomes. This association with CD is corroborated by the effective treatment of patients with active CD using starvation therapy, which probably decreases excessive inflammasome activation and prevents pyroptosis by promoting autophagic flux.

Furthermore, we demonstrated that the blockade of IL-1β ameliorates DSS-induced colitis in VAD mice. In contrast, the blockade of NLRP3 did not impact the severity of colitis, while the inhibition of mTOR resulted in the exacerbation of DSS-induced colitis in VAD mice. Based on these observations, increased IL-1β in the colonic LP due to non-canonical inflammasome activation could explain the exacerbation of DSS-induced colitis in VAD mice. Therefore, we believe the blockade of IL-1β can be used to treat certain types of colitis and warrants further exploration in clinical settings.

However, this study is not without limitations. First, inflammasomes have predominantly been examined in macrophages, although other myeloid cells, including dendritic cells and neutrophils [65,66], have also been investigated. We only focused on macrophages in the current study. Second, we used female mice for the DSS-induced colitis model, as they showed lesser individual differences in the volume of drinking water compared with male mice. Additionally, owing to an unstable supply of VAD SCID mice and their high lethality rate of DSS-induced colitis, we could not take body weight measurements up to day 5. Third, RAW 264.7 cells are derived from a tumor in a male mouse challenged with the Abelson murine leukemia virus, and the suppression of vitamin A levels in vivo involves not only RAR signaling but also RXR signaling. Therefore, we believe there would be a difference in activity between the RAR antagonist in the macrophage cell line and the suppression of vitamin A levels in this experimental system. Fourth, the analysis using VAD and VAS PECs pretreated with the RAR antagonist yielded results contrary to those obtained from RAW 264.7 cells, which may fail to truly represent the LPMC population found in inflamed tissues of patients with IBD. Fifth, we could not show higher levels of the active, cleaved form of GSDMD in RAR antagonist-pretreated RAW 264.7 cells. To propose a mechanism of IL-1β secretion through GSDMD-induced pore formation, the active form of GSDMD must be detected in cell supernatants. Sixth, the increased inflammation in the colitis model might be due to disrupted barrier functions in mice with VAD. Therefore, the effect of VAD on the epithelium and microbiome should be examined.

More recently, growing evidence has suggested the existence of unknown LPS response systems and starvation-sensing systems. Our findings suggest that the RAR and the respective signaling pathways regulate LPS via the fusion of autophagosomes and lysosomes in single-celled eukaryotes. Herein, we identified one mechanism underlying the retinoic acid-mediated regulation of inflammasome activation. Furthermore, certain intractable diseases, such as adult-onset Still’s disease (ASD) [67,68], interstitial lung disease (ILD) with clinically amyopathic dermatomyositis (CADM) [69,70,71], and intestinal Behçet’s disease (BD) accompanied with myelodysplastic syndrome (MDS) involving trisomy 8 [72], could be associated with macrophage activation syndrome (MAS), inflammasome activation, and pyroptosis. Cytokine storms, particularly MAS, may be involved in coronavirus disease 2019 (COVID-19)-associated pneumonia and its exacerbation [73]. Based on our data, vitamin A supplementation might be essential for IBD and probably for other auto-inflammatory diseases, such as FMF, ASD, ILD with CADM, intestinal BD accompanied by MDS involving trisomy 8, and severe COVID-19-associated pneumonia.

## 4. Materials and Methods

### 4.1. Preparation of VAD Mice

Pregnant C57BL/6J mice were purchased from Charles River (Yokohama, Japan), SCID (background strain is C57BL/6J) mice were acquired from Jackson Laboratories (Bar Harbor, ME, USA), and GFP-LC3#53 mice were obtained from the RIKEN BioResource Research Center (Tsukuba, Japan). Mice were fed a chemically defined diet lacking vitamin A (Oriental Yeast, Tokyo, Japan). Pups were weaned at 4 weeks of age and continuously maintained on the same diet until analysis at 10 weeks of age. VAD mice showed no signs of inanition and had equivalent body weights. Serum retinol levels of mice and humans were analyzed by BML (Tokyo, Japan), confirming that serum retinol levels of VAD mice were undetectable. VAS mice were fed a control diet containing retinol acetate. Mice were housed under specific-pathogen-free conditions in a temperature-controlled room at 22 °C under a 12 h light-dark cycle at the Institute for Animal Experimentation, Hirosaki University Graduate School of Medicine. All animal experiments were conducted in accordance with the Animal Research Ethics Committee of Hirosaki University Graduate School of Medicine and followed the Guidelines for Animal Experimentation of Hirosaki University (Permit number: M15007).

### 4.2. Induction of Colitis

Colitis was induced in female mice by feeding 2.5% (for SCID mice) or 4% (for C57BL/6J and GFP-LC3#53 mice) DSS (mol wt 5000; Wako Pure Chemical Co., Osaka, Japan) dissolved in drinking water (distilled water). Mice were followed up until day 10, as described previously [74]. We used 2.5% DSS to induce colitis in C57BL/6J SCID mice because of its high lethality rate in VAD SCID mice. VAD B6 mice were intraperitoneally administered 50 mg/head/day of mAB against IL-1β (Biolegend, San Diego, CA, USA) from day 0 (when DSS treatment was initiated) to day 3. Isotype-matched IgG was injected as a control [75]. A chemical NLRP3 inhibitor, MCC950 (MedChem Express, Monmouth Junction, NJ, USA), was intraperitoneally administered to mice (10 mg/kg) at day 0 and every two days thereafter [76]. An autophagy activator (a potent and specific mTOR inhibitor), rapamycin (MedChem Express), was dissolved in ethanol and diluted ≥20-fold with 5% Tween 80 (Sigma-Aldrich, St. Louis, MO, USA) solution containing 5% PEG-400 (Wako Pure Chemical Co.). The drug was administered intraperitoneally to mice (10 mg/kg) daily from days 0 to 7 [77]. The inhibitors, antagonists, or antibody treatments were co-administered with DSS. We observed no interactions between DSS treatment and other drugs.

### 4.3. Isolation of Colonic LP

Intestinal epithelial cells were purified as previously described [78]. In brief, mice were anesthetized by inhalation of isoflurane, followed by an abdominal incision. The thoracic cavity was opened and perfused through the left ventricle with 15 mL of 30 mmol/L EDTA in Hanks’ balanced salt solution (HBSS). At the end of perfusion, the entire colon, excluding the cecum, was removed, inverted, and placed in a cold tube with 2 mL of 2 mmol/L EDTA in cold HBSS. The tube was shaken using a minibeat beater, and the crypts in the supernatant were collected as the epithelium fraction. The tissue remnants were collected and used as the colonic LP fraction.

### 4.4. Western Blotting and Caspase Activity

Freshly isolated colonic tissues or macrophage cells were lysed in lysis buffer containing 1000 mM Tris-HCl, 500 mM EDTA, 0.5% Nonidet P-40 (Nakalai Tesque, Kyoto, Japan), 10% glycerol, 500 mM phenylmethylsulfonyl fluoride (Sigma-Aldrich, St. Louis, MO, USA), 100 mM sodium fluoride (Nakalai Tesque, Kyoto, Japan), 100 mM sodium vanadate (Sigma-Aldrich), 500 mM sodium pyrophosphate (Sigma-Aldrich, St. Louis, MO, USA), and protease inhibitor cocktail (Roche Molecular Biochemicals, Mannheim, Switzerland) at pH 6.8, followed by a 30 min incubation on ice. After centrifugation at 15,000× *g* and 4 °C, supernatants were collected as protein extract. Protein levels were determined using the BCA Protein Assay Kit (Thermo Fisher Scientific, Waltham, MA, USA). Cytosolic extracts (10 µg) were fractionated by 4–15% sodium dodecyl sulfate-polyacrylamide gel electrophoresis and electrotransferred to a PVDF membrane. Protein expression was detected by Western blotting with anti-caspase-11 primary antibody (1:1000) (ab 180673; Abcam), anti-GSDMD primary antibody (1:100) (ab 209845; Abcam), anti-NLRP3 primary antibody (1:1000) (AG-20B-0014; AdipoGen, Basel, Switzerland), anti-ASC primary antibody (1:1000) (sc-22514; Santa Cruz), anti-caspase-1 primary antibody (1:500) (sc-514; Santa Cruz), anti-IL-1β primary antibody (1:1000) (MAB401; R&D systems Inc., Minneapolis, MN, USA), anti-α-tubulin antibody (1:1000) (21255; Cell Signaling), and horseradish peroxidase-conjugated secondary antibody (1:5000–10,000) (R&D systems Inc., Minneapolis, MN, USA). The visualization of blots was performed using a chemiluminescent substrate. Caspase-1 activity in the purified LP fraction was measured using a colorimetric protease assay kit (BioVision Research Products, Mountain View, CA, USA), as described previously [74].

### 4.5. Bio-Plex

Cytokine assays were performed using a Bio-Plex^TM^ Mouse Cytokine 23-Plex Panel (1 × 96-well) (BioRad Laboratories, Inc., Hercules, CA, USA) according to the manufacturer’s instructions. Bio-Plex cytokine assays are multiplex bead-based assay systems that utilize Luminex-based technology [79].

### 4.6. Histological Assessment of Colon Sections

Paraffin-embedded sections of colonic tissues were cut and stained with hematoxylin and eosin (H&E). Histological severity was graded on a scale of 0 to 3 (0, normal; 1, slight; 2, moderate; 3, severe), considering cells infiltrating into LP, the appearance of erosions, reduction in crypts and glands, and height of epithelium, as previously described [80].

Tissue samples for GFP observation were prepared as follows: To prevent induction of autophagy during tissue preparation, GFP-LC3#53 mice were anesthetized using isoflurane inhalation, followed by immediate fixation by perfusion through the left ventricle with 4% paraformaldehyde (PFA) in 0.1 M sodium phosphate buffer (PB; pH 7.4). Tissues were harvested and further fixed with the same fixative for at least 4 h, followed by treatment with 15% sucrose in phosphate-buffered saline (PBS) for 4 h and then with 30% sucrose solution overnight. Tissue samples were embedded in Tissue-Tek OCT compound (Sakura Finetechnical Co., Ltd., Tokyo, Japan) and stored at −70 °C. The samples were sectioned into 5–7 µm-thick slices using a cryostat (CryoStar^TM^ NX70 Cryostat, Thermo Fisher Scientific), air-dried for 30 min, and stored at −20 °C until use. Fluorescent signals were observed under a fluorescence microscope. Electron microscopic analysis was performed by Filgen Inc. (Nagoya, Japan). For transmission electron microscopy (TEM), tissue samples were fixed in phosphate-buffered 2% glutaraldehyde and subsequently post-fixed in 2% osmium tetra-oxide for 2 h in an ice bath. Then, the specimens were dehydrated in graded ethanol and embedded in epoxy resin. The ultramicrotome technique was used to obtain ultrathin sections. Ultrathin sections stained with uranyl acetate for 10 min and lead staining solution for 5 min were subjected to TEM (JEM-1200 EX, JEOL).

### 4.7. Cell Culture

Macrophage cell line RAW 264.7 cells (ATCC TIB-71, Teddington, UK) were cultured in Dulbecco’s Modified Eagle’s Medium supplemented with 10% fetal bovine serum (FBS; Thermo Fisher Scientific) and seeded in 24-well culture plates at 1 × 10^6^ cells/well. Cells were treated with 10 mM Ro41-5253 (Abcam, Cambridgeshire, UK), as RAR inhibitor, or with dimethyl sulfoxide (DMSO; Sigma-Aldrich), as the control group, for 1 h, followed by simultaneous treatment with 10 µg/mL LPS (from *E. coli*: O111: B4, Sigma-Aldrich). The cultured supernatants and cells were collected 0, 1, 2, 4, 6, 12, and 24 h after LPS treatment. Cell viability was estimated by cell counting after staining with trypan blue (BioRad Laboratories). The titers of IL-1β, IL-18 (R&D Systems Inc., Minneapolis, MN, USA), and TNF-α (Thermo Fisher Scientific) in supernatants were measured using ELISA.

### 4.8. Cytotoxicity Assay

The LDH level was measured using CytoTox96 (non-radioactive cytotoxicity assay, Promega, Madison, WI, USA). To normalize for spontaneous lysis, the percentage of LDH release was calculated as follows: experimental LDH release (OD_490_)/maximum LDH release (OD_490_) × 100.

### 4.9. Flow Cytometric Analysis

To examine cell starvation, RAW 264.7 cells were washed with PBS twice and cultured in EBSS (Gibco) for 3 h. Mitochondrial ROS was measured in cells using MitoSOX (Invitrogen, Waltham, MA, USA) (5 µM for 15 min at 37 °C) [81,82]. Cells were washed with PBS, trypsinized and resuspended in PBS containing 1% heat-inactive FBS. Cells were analyzed using FACScan^TM^ with CELLQuest II^TM^ (Becton-Dickinson and Company, Franklin Lakes, NJ, USA).

### 4.10. Bacterial Strains, Plasmids, and Growth Conditions

*L. monocytogenes* 1b 1684 was grown in tryptic soy broth (BD Diagnosis Systems, Sparks, MD, USA) at 37 °C and stored at −80 °C until use [83]. pJEBAN6 plasmid encoding red fluorescence protein (DsRedEx) was transferred into *L. monocytogenes* by electroporation to obtain a red fluorescent stain [84]. *L. monocytogenes* harboring pJEBAN6 plasmid were cultured in tryptic soy broth supplemented with 5 µg/mL erythromycin (Wako Pure Chemical Industries, Osaka, Japan).

### 4.11. Infection of Mice with L. Monocytogenes

*L. monocytogenes* 1b 1684 and *L. monocytogenes* harboring pJEBAN6 plasmid were cultured in tryptic soy broth, and bacterial cells were prepared as previously described [85,86,87]. The concentration of washed cells was adjusted spectrophotometrically at 550 nm. Mice were infected intravenously with 0.2 mL of a solution containing 5 × 10^5^ CFU of viable *L. monocytogenes* in 0.01 M PBS (pH 7.4). The spleens and livers were harvested 24 and 48 h post-infection. The organs were homogenized in PBS with 1% 3-[(3-Cholamidopropyl) dimethylammonio] propanesulfonate (Wako Pure Chemical Industries Ltd.). The number of bacteria was counted by plating serial 10-fold dilutions of organ homogenates on tryptic soy agar (BD Diagnosis Systems). Colonies were routinely counted after 24 h [86].

### 4.12. Preparation of Splenic Macrophages and Bactericidal Activity

The phagocytic and bactericidal activities of splenic macrophages were determined as described previously [88]. Splenic macrophages were prepared by adhering spleen cells twice in RPMI 1640 medium (Thermo Fisher Scientific) supplemented with 10% FBS in a petri dish for 1 h at 37 °C under 5% CO_2_ conditions. Splenic macrophages, resuspended in antibiotic-free RPMI 1640 medium supplemented with 10% FBS at a concentration of 10^6^ cells/mL, were infected with *L. monocytogenes* at MOI of 10, followed by incubation at 37 °C for 30 min. Subsequently, macrophages were washed three times with RPMI 1640 medium supplemented with 10% FBS and 50 µg/mL gentamicin and were transferred into a 96-well flat-bottom microplate (Nunc, Roskilde, Denmark) at a density of 10^6^ cells in 100 µL per well. The infected cells were lysed by treatment with RPMI 1640 medium containing 1% (wt/vol) CHAPS at 0, 2, 4, and 6 h. Lysates from three wells were pooled, and the number of viable intracellular bacteria in each sample was determined by culturing on a tryptic soy agar plate.

### 4.13. Statistical Analysis

Data were expressed as mean ± standard error of the mean. An unpaired *t*-test (GraphPad PRISM 7, GraphPad Software, Inc., San Diego, CA, USA) was used to determine the significant differences in obtained values. A *p*-value of <0.05 or 0.01 was considered significant. Each experiment was repeated at least twice.

## 5. Conclusions

In conclusion, this study uncovered novel mechanisms of vitamin A concerning innate immunity, inflammasome, and pyroptosis activity. The findings of the present study suggest that vitamin A plays an important role in the efficient fusion between autophagosomes and lysosomes and affords protective effects against colitis, thereby highlighting the potential of retinoic acid as a potential new agent to treat IBD. Given that the microbiome of VAD mice was not examined, a follow-up permeability and microbiome analysis of VAD mice should be conducted.

## Figures and Tables

**Figure 1 ijms-24-08684-f001:**
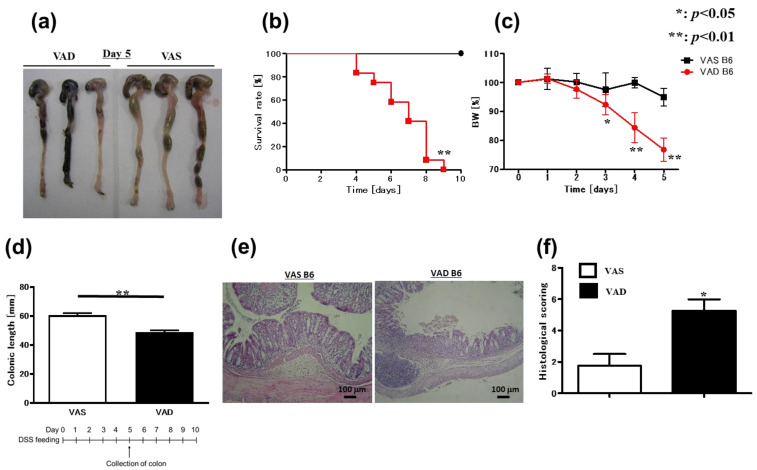
VAD B6 mice show more severe DSS-induced colitis than VAS mice. (**a**–**d**) VAS and VAD B6 mice were fed 4% DSS and were followed up until day 10. (**a**) Macroscopic findings (*n* = 3), (**b**) survival rate (*n* = 12), (**c**) body weight course (*n* = 8), (**d**) colonic length (*n* = 6), (**e**) H&E staining, and (**f**) histological scoring (*n* = 4) are shown. In (**b**), data were analyzed using the Kaplan–Meier test. In (**c**,**d**,**f**), data are presented as mean ± SEM (*n* > 2 independent experiments). An unpaired *t*-test was used for statistical analysis (* *p* < 0.05, ** *p* < 0.01). DSS, dextran sulfate sodium; H&E, hematoxylin and eosin; SEM, standard error of the mean; VAD, vitamin A-deficient; VAS, vitamin A-sufficient.

**Figure 2 ijms-24-08684-f002:**
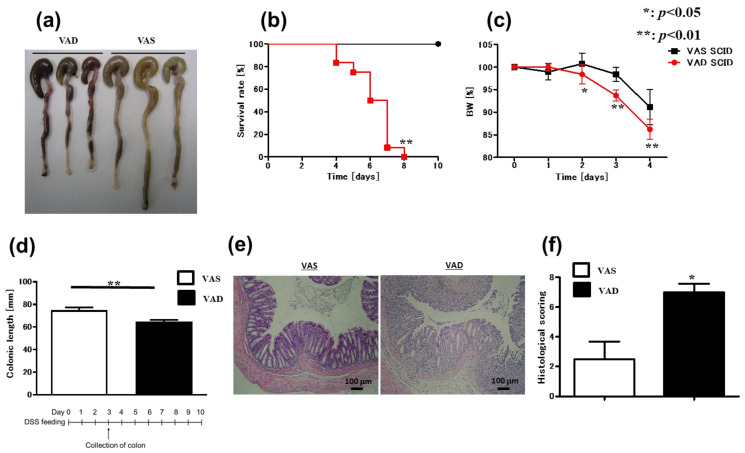
VAD SCID mice exhibit more severe DSS-induced colitis than VAS mice. (**a**–**d**) VAS and VAD SCID mice were fed 2.5% DSS and were followed up until day 10. (**a**) Macroscopic findings (*n* = 3), (**b**) survival rate (*n* = 9), (**c**) body weight course (*n* = 8), (**d**) colonic length (*n* = 6), (**e**) H&E staining, and (**f**) histological scoring (*n* = 4) are shown. In (**b**), data were analyzed with the Kaplan–Meier test. In (**c**,**d**,**f**), data are presented as mean ± SEM (*n* > 2 independent experiments). An unpaired *t*-test was performed for statistical analysis (* *p* < 0.05, ** *p* < 0.01). DSS, dextran sulfate sodium; H&E, hematoxylin and eosin; SCID, severe combined immunodeficient; SEM, standard error of the mean; VAD, vitamin A-deficient; VAS, vitamin A-sufficient.

**Figure 3 ijms-24-08684-f003:**
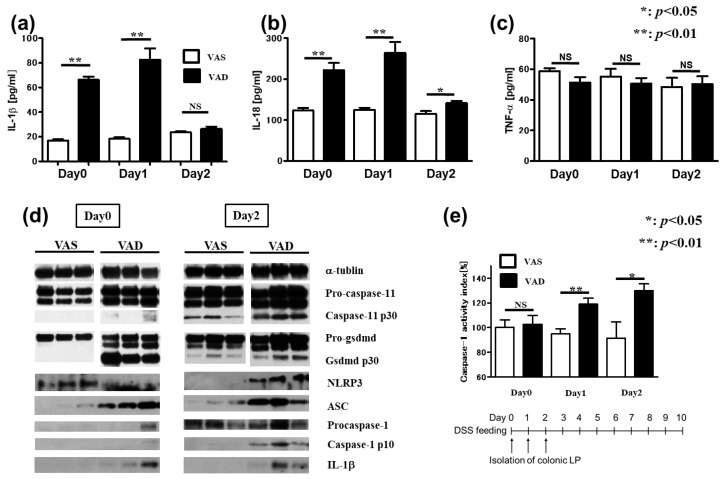
Inflammasome-related proteins increase in colonic lamina propria of VAD mice. (**a**–**c**) Profile of cytokine production in lamina propria (LP) from DSS-treated VAD mice using a multiplex system (Bio-Plex, Bio-Rad) (*n* = 5). VAS and VAD mice were fed with DSS in drinking water as described in the Materials and Methods. On day 0, 1, and 2 after DSS feeding, colonic LP of mice was isolated using a modified Bjerk’s method. The cytokine assay was performed using Bio-Plex multi-assay system (Bio-Rad Laboratories). (**d**) Immunoblotting for inflammasome-related proteins was performed using LP isolated from DSS-treated mice. (**e**) Caspase-1 activities in colonic LP were measured using a colorimetric protease assay kit (Bio Vision Research Product) (*n* = 5). Data with error bars are represented as means ± SEM (*n* > 2 independent experiments). An unpaired *t*-test was used for statistical analysis (NS, not significant; * *p* < 0.05, ** *p* < 0.01). Each panel represents an experiment performed at least three times. DSS, dextran sulfate sodium; SEM, standard error of the mean; VAD, vitamin A-deficient.

**Figure 4 ijms-24-08684-f004:**
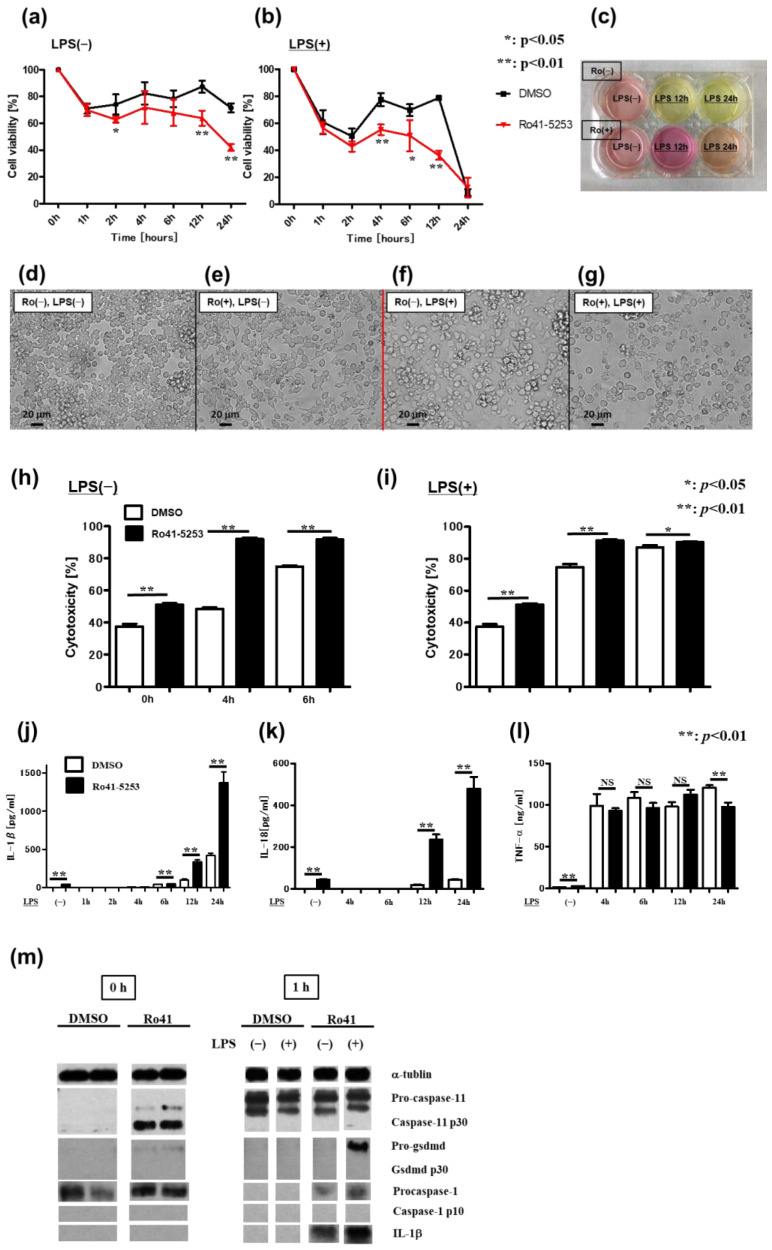
Macrophages pretreated with RAR antagonist exhibit increased pyroptosis mediated via non-canonical inflammasome signaling. (**a**–**m**) Effect of RAR antagonist (Ro41-5253) on cell viability of murine macrophage cells (RAW 264.7) treated with LPS. RAW 264.7 cells were cultured in 10 mg/mL LPS for 24 h, pretreated with or without 10 mM Ro41-5253 for 1 h. Cell viability, cytotoxicity, cytokine production, and immunoblotting were examined at each time point. (**a**) Cell viability in the absence of LPS (*n* = 4), (**b**) cell viability in the presence of 10 mg/mL LPS (*n* = 4), (**c**) representative images of supernatants, and (**d**–**g**) morphology of RAW 264.7 cells upon Ro41-5253 and LPS treatment. (**h**,**i**) LDH was measured using Cytotox96 (non-radioactive cytotoxicity assay, Promega). (**h**) Cytotoxicity assay without LPS (*n* = 6) and (**i**) cytotoxicity assay in the presence of 10 mg/mL LPS (*n* = 6). (**j**–**l**) Titers of IL-1β, IL-18, and TNF-α in supernatants were measured using ELISA. (**j**) IL-1β production (*n* = 4), (**k**) IL-18 production (*n* = 4), (**l**) TNF-α production (*n* = 4), and (**m**) immunoblotting of RAW 264.7 cells upon Ro41-5253 and LPS treatment. Graphs show the mean ± SEM of duplicate wells and represent three independent experiments. An unpaired *t*-test was used for statistical analysis (NS, not significant; * *p* < 0.05, ** *p* < 0.01). Each panel represents an experiment performed at least three times. IL-1β, -18, interleukin-1β, -18; LDH, lactate dehydrogenase; LPS, lipopolysaccharide; RAR, retinoic acid receptor; TNF-α, tumor necrosis factor-α.

**Figure 5 ijms-24-08684-f005:**
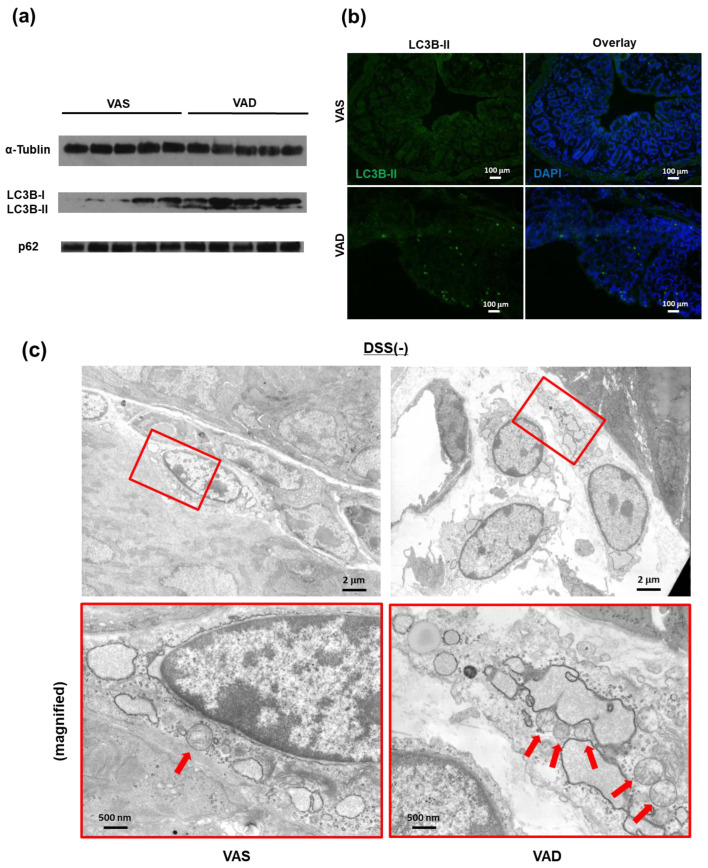
VAD decreases autophagic flux by reducing the fusion of autophagolysosomes. (**a**) Immunoblotting for autophagy-related proteins was performed using isolated colonic lamina propria (LP) of B6 mice (day 2). (**b**) Autophagy in colonic tissues of GFP-LC3 transgenic mice (GFP-LC3#53) (day 1). (**c**) Electron microscopic analysis of colonic LP with or without DSS treatment (day 1). The areas of red box were magnified. The red arrows indicate mitochondria. (**d**,**e**) Effects of RAR antagonist (Ro41-5253) on autophagy of murine macrophage cells (RAW 264.7). (**d**) Immunoblotting of autophagy-related proteins. RAW 264.7 cells were pretreated with or without 10 mM Ro41-5253 for 1 h and then harvested in the presence of 10 mg/mL LPS after 24 h. (**e**) Mitochondrial ROS was measured by MitoSOX staining after 4 h of EBSS treatment. Each panel represents an experiment performed at least three times. EBBSS, Earle’s Balanced Salt Solution; LPS, lipopolysaccharide; RAR, retinoic acid receptor; ROS, reactive oxygen species. #### indicates a value greater than 9999.

**Figure 6 ijms-24-08684-f006:**
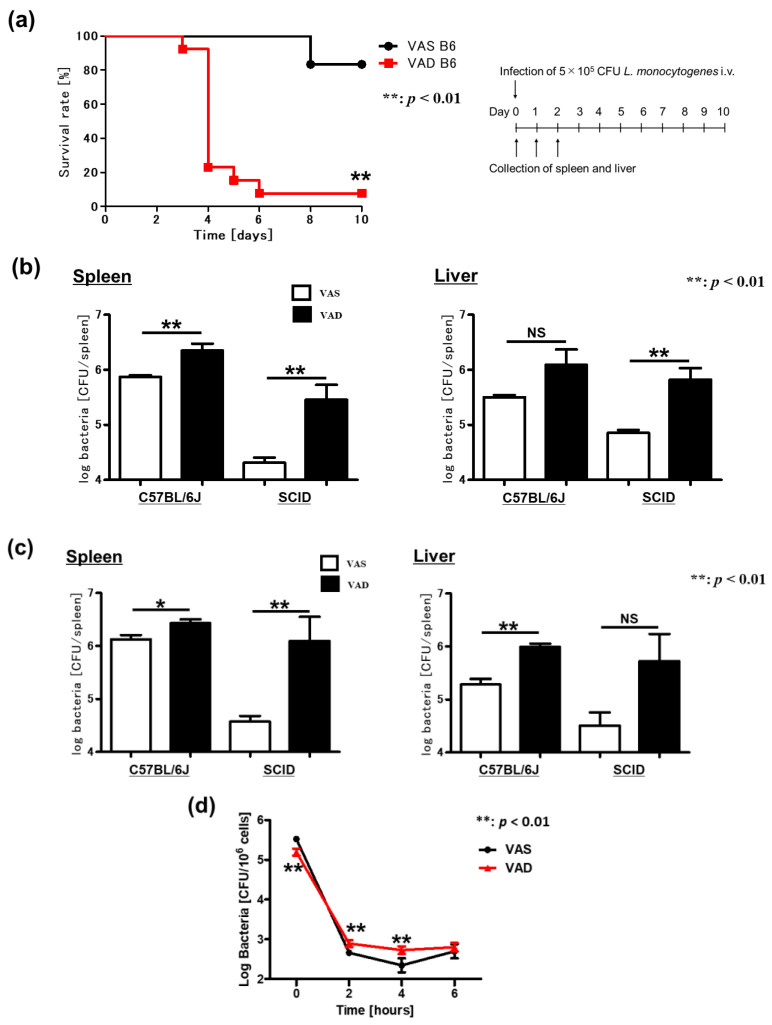
VAD impairs autophagic and listericidal activities of macrophages and host resistance to *Listeria monocytogenes* infection. (**a**–**c**) C57BL/6J and SCID mice were intravenously infected with 5 × 10^5^ CFU of *L. monocytogenes*. Mice were followed up until day 10, or spleens and livers were harvested 24 h (day 1) and 48 h (day 2) post-infection. (**a**) Survival rate (*n* = 12), (**b**) the number of bacteria in organs (day 1, *n* = 4), (**c**) the number of bacteria in organs (day 2, *n* = 4), and (**d**) splenic macrophages of VAS and VAD mice infected with *L. monocytogenes* at the multiplicity of infection (MOI) of 10. The numbers of viable bacteria in cell lysates were counted at each time point (*n* = 5). (**e**) Fluorescence microscopic findings of GFP-LC3 transgenic mice (GFP-LC3#53) infected with DsRedEx-labeled *L. monocytogenes*. Mice were intravenously infected with 5 × 10^5^ CFU of *L. monocytogenes*. The livers were removed 48 h (day 2) post-infection. After fixation by perfusion with 4% paraformaldehyde (PFA), fluorescent signals were observed under a fluorescence microscope. (**f**) Electron microscopic findings of mice infected with *L. monocytogenes*. Mice were intravenously infected with 5 × 10^5^ CFU of *L. monocytogenes*. The spleens were harvested pre-infection (day 0) and 24 h (day 1) post-infection. The red circles indicate autophagosomes. In (**a**), data were analyzed using the Kaplan–Meier test. In (**b**–**d**), data are presented as mean ± SEM (*n* > 2 independent experiments). An unpaired *t*-test was used for statistical analysis (NS, not significant; * *p* < 0.05, ** *p* < 0.01). Each panel represents an experiment performed at least three times. CFU, colony-forming unit; SEM, standard error of the mean; VAD, vitamin A-deficient; VAS, vitamin A-sufficient.

**Figure 7 ijms-24-08684-f007:**
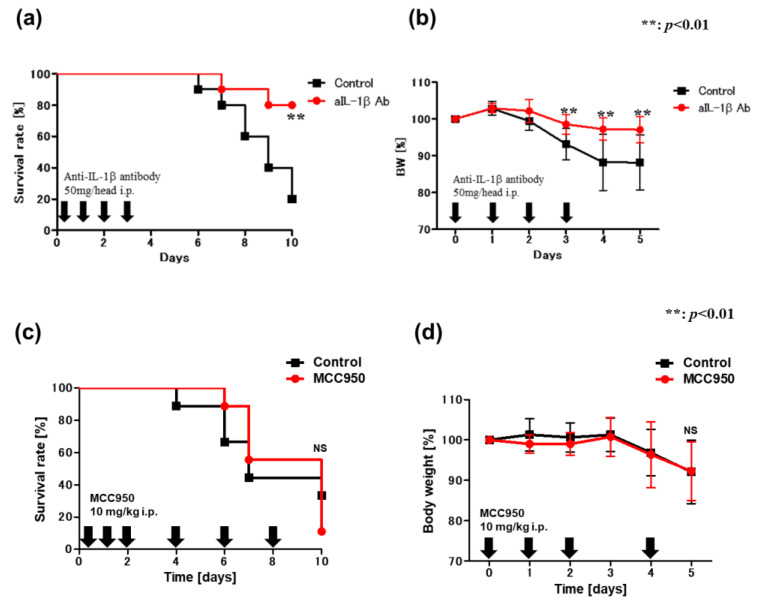
Blockade of IL-1β ameliorates DSS-induced colitis in VAD mice. (**a**,**b**) Effect of monoclonal antibody against IL-1β on DSS-induced colitis in VAD B6 mice. VAD B6 mice were injected intraperitoneally with 50 mg/head/day of monoclonal antibody against IL-1β (BioLegend) from day 0 (when DSS treatment was initiated) to day 3. Isotype-matched IgG was injected as a control. (**a**) Survival rate, (**b**) body weight course. (**c**,**d**) Survival rate and body weight course after MCC950 administration. (**e**,**f**) Survival rate and body weight course after rapamycin administration. In (**a**,**c**,**e**), data were analyzed using the Kaplan–Meier test. In (**b**), data are presented as mean ± SEM (*n* > 2 independent experiments). An unpaired *t*-test test was used for statistical analysis (NS, not significant; ** *p* < 0.01). DSS, dextran sulfate sodium; IL-1β, interleukin-1β; SEM, standard error of the mean; VAD, vitamin A-deficient.

## Data Availability

The data presented in this study are available on request from the corresponding author. The data are not publicly available due to privacy and ethical restrictions.

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
