# Peer review of "Vitamin A Promotes the Fusion of Autophagolysosomes and Prevents Excessive Inflammasome Activation in Dextran Sulfate Sodium-Induced Colitis"

_ijms, 2023, doi:10.3390/ijms24108684_

Round 1

Reviewer 1 Report

In this study, the authors investigate the role of vitamin A, and more specifically of retinoic acid, in the regulation of intestinal inflammation through inflammasome signaling. For this, the authors carry out a study with a high experimentability, using a model of DSS-colitis in C57BL/6J Wild-type and SCID mice, fed for 6 weeks from weaning, with diets without vitamin A or with this vitamin. Thus, the researchers were able to verify that vitamin A deficiency caused an increase in the intestinal production of IL1b and Il18 before administering DSS, while after DSS the colitis was more severe and independent of T and B cells. Dietary vitamin A restriction increases the expression of inflammasome-related proteins in the lamina propria of the colon and produces mitochondrial alterations related to a decrease in autophagy in mononuclear cells. On the other hand, the authors suggest the existence of inflammasome independent pyroptosis using a retinoic acid receptor (RAR) antagonist (both in the presence or not of LPS) in vitro assays with RAW264.7 macrophages. Likewise, in the same in vitro experimental system as the previous one, complemented with an in vivo model with GFP-LC3 transgenic mice, the authors described the reduced fusion of autophagosomes and lysosomes by the RAR antagonist or due to vitamin A deficiency in the diet, respectively. Likewise, in the same in vitro experimental system as the previous one, complemented with an in vivo model of GFP-LC3 transgenic mice, the authors described the reduced fusion of autophagosomes and lysosomes against the RAR antagonist or due to vitamin A deficiency in the diet. respectively. The consequences of this autophagic deficit due to lack of vitamin A impair the action of macrophages against Listeria monocytogenes infection. However, the authors reversed the severity of DSS colitis in mice fed a diet without vitamin A by treating it with anti-IL1b. In short, a study that tries to decipher the mechanisms by which vitamin A can protect against the development of intestinal inflammatory processes such as those that develop in IBD. The study is interesting and can provide new knowledge. However, even though in the discussion the authors talk about some limitations of their study, which is to be appreciated, there are other details that should be improved.

In my opinion, the description of the experimental design, which is currently confusing, should be improved. It is not clear how many days the DSS is applied for the induction of colitis, the total number of animals and how they are distributed is not mentioned (mentioning the "n" in the figures does not clarify the design). Were the mice used male or female? Are inhibitors, antagonists, or antibody treatments co-administered with DSS? When is the evaluation done before the DSS? The authors must justify why there are body weight measurements taken up to day 4 and others up to day 5 (Figure 1 and 2, c). In Figure 3, are days 0, 1, and 3 shown on the graphs before or after DSS? Also, in material and methods, colitis was induced with 2.5% or 4% DSS, but only 4% is mentioned in the article.

Other comments:

It would be interesting for the authors to mention if there are differences in the evaluations carried out between Wild-type and SCID mice. It is curious to observe how the macroscopic and microscopic evaluations are similar between both genetic strains, both in a diet with or without vitamin A.

It would also be desirable to mention why C57BL/6J mice have been chosen for DSS induction of colitis. In addition to justifying the molecular weight of the DSS used in the study.

The content in lines 234-236 should have been mentioned in section 2.3.

The use of the RAR antagonist in macrophage cultures leads to an increase in pyroptosis via non-canonical inflammasome. Would there be differences in activity between the RAR antagonist and the suppression of vitamin A in this experimental system?

Delete (d,e) from line 273.

Line 116 shows "female B6 mice". What does it mean?

In the graphs of figure 6 represented with bars, the meaning of the white and black bars must be added, as has been done in the rest of the figures.

Again, in figure 7 discrepancies appear in the representation of the days in the graphs. Furthermore, the days for the administration of anti-IL1b and rapamycin are not well specified, why are they different? By the way, there are some repeated graphs. However, the result obtained with the NLRP3 inhibitor, MCC950, is not shown.

The authors propose the potential of anti-IL1 as a therapy to reverse colitis in mice fed diets without vitamin A. But would it be plausible to think of reversing the severity of DSS colitis in this scenario by applying vitamin A or refed with a normovitamin diet?

It would be interesting to add the formulation of the diets used in this study, even if it is in the supplementary data.

Correct "DD" in line 448.

line 610, specify the time-points to which the authors refer.

I sincerely hope that the comments are useful for the authors and for the improvement of this interesting study.

Author Response

Response to Reviewers’ comments

              The manuscript has been revised and all of the reviewers’ comments have been addressed. In addition, we submitted high-resolution (300dpi) versions of all figures in separate files. Our responses to the reviewers’ comments are as follows:

・Response to Reviewer #1: Thank you very much for providing these valuable suggestions. We agree with your advice and have revised the manuscript accordingly to reflect the corresponding suggestions (red color).

  1. It is not clear how many days the DSS is applied for the induction of colitis, the total number of animals and how they are distributed is not mentioned.

We agree with you. Colitis was induced by continuously feeding DSS dissolved in the drinking water, and mice were followed up until day 10 to assess the survival rate. We added this description to the Materials and Methods section. Unfortunately, because of an unstable supply of VAD mice and the high lethality rate of DSS-induced colitis in VAD mice, we had to limit the number of animals distributed in the indicated groups.

  1. Were the mice used male or female?

We used female mice for the DSS-induced colitis model, as they showed lesser individual differences in the volume of drinking water compared with male mice. We added a corresponding description to the Materials and Methods section.

  1. Are inhibitors, antagonists, or antibody treatments co-administered with DSS?

The inhibitors, antagonists, or antibody treatments were co-administered with DSS. We observed no interactions between DSS treatment and other drugs, as all drugs were intraperitoneally administered.

  1. When is the evaluation done before the DSS?

The evaluation before DSS treatment was done for the “day 0” group. “Day 0” refers to the day on which DSS treatment was initiated.

  1. The authors must justify why there are body weight measurements taken up to day 4 and others up to day 5 (Figure 1 and 2, c).

We could not take body weight measurements up to day 5 owing to the high lethality rate of DSS-induced colitis in VAD SCID mice, as mentioned above.

  1. In Figure 3, are days 0, 1, and 3 shown on the graphs before or after DSS?

As described in the Materials and Methods section, “day 0” refers to the day on which DSS treatment was initiated. Therefore, days 0, 1, and 2 shown in the graphs in Figure 3 are after DSS treatment.

  1. Also, in material and methods, colitis was induced with 2.5% or 4% DSS, but only 4% is mentioned in the article.

As you pointed out, the description of the experimental design was confusing. We added the relevant description to both the Materials and Methods and Results sections.

  1. It would be interesting for the authors to mention if there are differences in the evaluations carried out between Wild-type and SCID mice.

We used 2.5% DSS to induce colitis in C57BL/6J SCID mice because of the high lethality rate of DSS-induced colitis in VAD SCID mice. We added the description to the Materials and Methods section.

  1. It would also be desirable to mention why C57BL/6J mice have been chosen for DSS induction of colitis.

We chose C57BL/6J mice to define the effect of vitamin A deficiency on murine experimental colitis because they are more susceptible to DSS-induced colitis compared with the other strains (e.g., Balb/c mice).

  1. In addition to justifying the molecular weight of the DSS used in the study.

We used DSS with a molecular weight of 5000 in the study and described this in the Materials and Methods section.

  1. The content in lines 234-236 should have been mentioned in section 2.3.

The description in lines 234–236 is a part of the legend of Figure 4 and subtitle 2.4. Therefore, please specify in more detail about the clarification needed in section 2.3.

  1. Would there be differences in activity between the RAR antagonist and the suppression of vitamin A in this experimental system?

RAW264.7 cells are derived from a tumor in a male mouse challenged with the Abelson murine leukemia virus, and the suppression of vitamin A levels in vivo involves not only RAR signaling but also RXR signaling. Therefore, we think there would be difference in activity between the RAR antagonist in the macrophage cell line and the suppression of vitamin A levels in this experimental system.

  1. Delete (d,e) from line 273.

We deleted (d, e) from line 273.

  1. Line 116 shows "female B6 mice". What does it mean?

As you mentioned, the description was confusing. We modified the Materials and Methods section and deleted “female” from the legends of Figures 1 and 2.

  1. In the graphs of figure 6 represented with bars, the meaning of the white and black bars must be added, as has been done in the rest of the figures.

We added the meaning of the white (VAS) and black (VAD) bars in the graphs presented in Figure 6 (b, c).

  1. Furthermore, the days for the administration of anti-IL1b and rapamycin are not well specified, why are they different?

The methods for administration of a monoclonal antibody against IL-1b, MCC950, and rapamycin were based on the protocols described in the following papers.

IL-1b mAb: Williams, R.O.; Marinova-Mutafchieva, L.; Feldmann, M.; Maini, R.N. Evaluation of TNF-alpha and IL-1 blockade in collagen-induced arthritis and comparison with combined anti-TNF-alpha/anti-CD4 therapy. Immunol. 2000, 165, 7240–7245. DOI: 10.4049/jimmunol.165.12.7240

MCC950: Coll, R.C.; Robertson, A.A.B.; Chae, J.J.; Higgins, S.C.; Munoz-Pilaillo, R.; Inserra, M.C.; Vetter, I.; Dungan, L.S.; Monks, B.G.; Stutz, A.; Croker, D.E.; Butler, M.S.; Haneklaus, M.; Sutton, C.E.; Nunez, G.; Latz, E.; Kastner, D.L.; Mills, K.H.G.; Masters, S.L.; Schroder, K.; Cooper, M.A.; O’Neill, L.A. A small-molecule inhibitor of the NLRP3 inflammasome for the treatment of inflammatory diseases. Nat. Med. 2015, 21, 248–255. DOI: 10.1038/nm.3806

Rapamycin: Pupyshev, A.B.; Tikhonova, M.A.; Akopyan, A.A.; Tenditnik, M.V.; Dubrovina, N.I.; Korolenko, T.A. Therapeutic activation of autophagy by combined treatment with rapamycin and trehalose in a mouse MPTP-induced model of Parkinson's disease. Pharmacol. Biochem. Behav. 2019, 177, 1–11. DOI: 10.1016/j.pbb.2018.12.005

  1. By the way, there are some repeated graphs. However, the result obtained with the NLRP3 inhibitor, MCC950, is not shown.

Figure 7 (c, d) was the same as Figure 7 (a, b) and this was a mistake. We changed Figure 7 (c, d) to the correct images.

  1. Would it be plausible to think of reversing the severity of DSS colitis in this scenario by applying vitamin A or refed with normovitamin diet?

This point has already been considered and the experiments have been conducted. A paper containing these data is under the submission process. We hope you will understand our situation that this data cannot be added to this manuscript.

  1. It would be interesting to add the formulation of the diets used in this study, even if it is in the supplementary data.

We used the modified AIN-93 from Oriental Yeast, Co Ltd., Tokyo, Japan. The formulation of AIN-93G and AIN-93M was found from the following website. You can follow this formula by modifying the AIN93 vitamin mixture with or without vitamin A supplementation. The name of each ingredient (in English) is also found at DOI:10.21767/2171-6625.1000234.

https://www.clea-japan.com/products/special_diet/item_d0220

  1. Correct "DD" in line 448.

We corrected “DD” in line 448 to “DSS.”

  1. line 610, specify the time-points to which the authors refer.

As you pointed out, we added the specific time points to the Materials and Methods section.

We believe that incorporating your suggestions into the revised version has improved the manuscript. Thank you once again.

Reviewer 2 Report

This paper described vitamin A function from the viewpoint of autophagolysomes and inflammasome of macrophages using a murine experimental model of dextran sulfate sodium-induced colitis (DSS colitis).

 The research is almost well analyzed and organized, and the results are reasonable.

There are several minor points to be revised before the acceptance, as follows.

1.      Scale bars should be added to the histologic and electron microscopic (EM) figures for readers' reference.

2.      Fig 5. (c): The contrast of EM photographs should be more precise. The difference in mitochondria is not fine between VAD and VAS.

3.      Fig 7. (c, d): They are the same as (a, b). They should be changed to real figures.

4.      Line 403: The reference [36] an original DSS colitis model paper, is not appropriate here.  Instead, Okayasu I et al. Vitamin A inhibits development of dextran sulfate sodium-induced colitis and colon cancer in a mouse model. BioMed Res Int, Vol 216. Article ID 4874809. http://dx.doi.org/10.1155/2016/4874809 would be appropriate.

5.      Line 426 and 446, (data not shown): If you have data, please add the results in the supplementary files.

6.      Line 504 and follow Line 514. This seems to be overexpressed and too much speculated. The authors need to add references supporting it. For example, low concentration of vitamin A in the blood in patients with those diseases.

Author Response

Response to Reviewers’ comments

              The manuscript has been revised and all of the reviewers’ comments have been addressed. In addition, we submitted high-resolution (300dpi) versions of all figures in separate files. Our responses to the reviewers’ comments are as follows:

Response to Reviewer #2: Thank you very much for providing these valuable suggestions. We agree with all six pieces of advice and revised the manuscript accordingly to reflect these suggestions (light blue color).

  1. Scale bars should be added to the histologic and electron microscopic (EM) figures for readers' reference.

We agree with you. Scale bars have been added to the histologic figures (Figures 1e, 2e, 5b, 6e) and electron microscopy-based figures (Figures 5c, 6f).

  1. Fig 5. (c): The contrast of EM photographs should be more precise.

We replaced the EM photographs with those of higher definition. In addition, we submitted high-resolution (300dpi) versions of figures in separate files.

  1. Fig 7. (c, d): They are the same as (a, b). They should be changed to real figures.

As you pointed out, Figures 7 (c, d) were the same as Figures 7 (a, b), and this was by mistake. We changed Figures 7 (c, d) to the correct images.

  1. Line 403: The reference [36] an original DSS colitis model paper, is not appropriate here. Instead, Okayasu I et al. Vitamin A inhibits development of dextran sulfate sodium-induced colitis and colon cancer in a mouse model. BioMed Res Int, Vol 216. Article ID 4874809. http://dx.doi.org/10.1155/2016/4874809 would be appropriate.

Thank you very much for your kind advice. We changed the reference [36] to the new reference as recommended.

  1. Line 426 and 446, (data not shown): If you have data, please add the results in the supplementary files.

We used the “data not shown” description in lines 426 and 446 because these data are in the preliminary stage. We are conducting further experiments for reproducibility and are preparing another paper for submission. To avoid confusion, we decided to add the following sentences instead:

Lines 425–427:

“To support this hypothesis, further comparison of IL-18 expression in the epithelial fraction and the colonic LP of VAD B6 mice is required.”

Lines 444–446

“One possible mechanism underlying this association may be related to increased epithelial permeability in VAD mice, resulting in bacterial translocation into the colonic LP.”

  1. Line 504 and follow Line 514. This seems to be overexpressed and too much speculated.

We agree with you. We corrected the sentence in the Discussion part, to avoid exaggeration.

We believe that incorporating your suggestions into the revised version has improved the manuscript. Thank you once again.

Reviewer 3 Report

The manuscript entitled “Vitamin A Promotes the Fusion of Autophagolysosomes and Prevents Excessive Inflammasome Activation in Dextran Sulfate Sodium-Induced Colitis” describes the impact of vitamin A deficiency in the development of colitis in vivo and in vitro. Although the effect of vitamin A supplementation on mice subjected to DSS-induced colitis was already described. Here, the Authors show interesting novel data regarding the mechanism underlying vitamin A actions during intestinal inflammation. In general, the manuscript is well-written and organized. However, I would suggest the following amendments:

1.     After a very extensive introduction, the hypothesis of the study, the knowledge gap and the relevance of the study are not clearly defined. Given that the effects of vitamin A in the development of DSS-induced colitis are well documented, the Authors should highlight the novelty of their study, which is the impact of vitamin A on innate immunity, inflammasome and pyroptosis activity, and the new mechanisms described.

2.     The first time VAD appears in the main text should be in brackets.

3.     Detailed information on how DSS-induced colitis was performed should be included in methods (duration of DSS, DSS withdrawal?, etc)

4.     Consider deleting some semicolons across the text. In my opinion, there is an incorrect use of that punctuation mark.

Author Response

Response to Reviewers’ comments

              The manuscript has been revised and all of the reviewers’ comments have been addressed. In addition, we submitted high-resolution (300dpi) versions of all figures in separate files. Our responses to the reviewers’ comments are as follows:

Response to Reviewer #3: Thank you very much for providing these valuable suggestions. We agreed with all four pieces of advice and revised the manuscript accordingly to reflect these suggestions (green color).

  1. After a very extensive introduction, the hypothesis of the study, the knowledge gap and the relevance of the study are not clearly defined. Given that the effects of vitamin A in the development of DSS-induced colitis are well documented, the Authors should highlight the novelty of their study, which is the impact of vitamin A on innate immunity, inflammasome and pyroptosis activity, and the new mechanisms described.

              Thank you very much for your kind advise. We added a description regarding the novelty of the study to the Conclusion section.

  1. The first time VAD appears in the main text should be in brackets.

              We agree with you. We placed VAD in brackets the first time it appears in the main text.

  1. Detailed information on how DSS-induced colitis was performed should be included in methods (duration of DSS, DSS withdrawal?, etc)

              We added detailed information on how DSS-induced colitis was performed to the Materials and Methods section of the revised manuscript.

  1. Consider deleting some semicolons across the text. In my opinion, there is an incorrect use of that punctuation mark.

              The manuscript was re-checked by a native speaker concerning the correct use of semicolons and punctuation marks, and we corrected the manuscript accordingly.

We believe that incorporating your suggestions into the revised version has improved the manuscript. Thank you once again.

Round 2

Reviewer 1 Report

Dear Authors,

thank you very much for your answers. I think, it would be good to include in the discussion the justifications given in questions 2,3,5 and 12, or even in the material and methods section, as limitations of the study. I mean the exclusive use of females, or that the therapies are preventive when co-administering them with DSS, in addition to including that they do not have reactions with DSS. On the other hand, the answer to question 6 is not clear and, therefore, neither is the corresponding figure. Also, the authors have introduced reference 36 to justify part of the experimental design. This reference uses DSS of mol weight 54000. Why do you use DSS mol weight of 5000?, does it achieve the same colitis as with the higher mol weight?, this detail should be justified in the article. Finally, I consider that the authors should incorporate a scheme with the experimental design to facilitate the  result interpretation. 

Author Response

Response to Reviewer’s comments (Round 2)

Response to Reviewer #1: Thank you very much for providing these valuable suggestions. We agree with your advice and have revised the manuscript accordingly to reflect the corresponding suggestions (red color).

  1. I think, it would be good to include in the discussion the justifications given in questions 2,3,5 and 12, or even in the material and methods section, as limitations of the study.

We agree with you. According to your suggestion, we added the justification of Question 2 (use of only female mice in this study), Question 5 (body weight measurement of DSS-induced colitis in VAD SCID mice up to day 4), and Question12 (differences in activity between the RAR antagonist and the suppression of vitamin A) in the Discussion section, as limitations of the study. Additionally, we added the justification of Question 3 (co-administration of inhibitors, antagonists, or antibody with DSS) in the Materials and Methods section.

  1. On the other hand, the answer to question 6 is not clear and, therefore, neither is the corresponding figure.

As you mentioned, the description was confusing. For Question 6, we added some sentences in the legend of Figure 3.

Figure 3. Inflammasome-related proteins increase in colonic lamina propria of VAD mice. (a-c) Profile of cytokine production in lamina propria (LP) from DSS-treated VAD mice using a multiplex system (Bio-Plex, Bio-Rad) (n=5). VAS and VAD mice were fed with DSS in drinking water as described in the Materials and Methods. On day 0, 1 and 2 after DSS feeding, colonic LP of DSS-treated mice was isolated using a modified Bjerk’s method. …

  1. Also, the authors have introduced reference 36 to justify part of the experimental design. This reference uses DSS of mol weight 54000. Why do you use DSS mol weight of 5000?, does it achieve the same colitis as with the higher mol weight?, this detail should be justified in the article.

Concerning molecular weight of DSS, reference 36 in the Materials and Methods section was mistake. We changed the reference from reference 36 to reference 41 (Sakuraba, H.; Ishiguro, Y.; Yamagata, K.; Munakata, A.; Nakane, A. Blockade of TGF-Beta Accelerates Mucosal Destruction Through Epithelial Cell Apoptosis. Biochem. Biophys. Res. Commun. 2007, 359, 406–412. DOI: 10.1016/j.bbrc.2007.05.117).

  1. Finally, I consider that the authors should incorporate a scheme with the experimental design to facilitate the result interpretation.

As you pointed out, we added each scheme with the experimental disign in Figure 1, 2, 3, and 6.

We believe that incorporating your suggestions into the revised version has improved the manuscript. Thank you once again.
